# Emergence of charge density waves and a pseudogap in single-layer TiTe$_2$

P. Chen [1,2,3], Woei Wu Pai[4,5], Y.-H. Chan[6], A. Takayama[7], C.-Z. Xu [1,2,3], A. Karn[4], S. Hasegawa[7], M.Y. Chou[5,6,8], S.-K. Mo [3], A.-V. Fedorov[3] & T.-C. Chiang[1,2,5]

Two-dimensional materials constitute a promising platform for developing nanoscale devices and systems. Their physical properties can be very different from those of the corresponding three-dimensional materials because of extreme quantum confinement and dimensional reduction. Here we report a study of TiTe$_2$ from the single-layer to the bulk limit. Using angle-resolved photoemission spectroscopy and scanning tunneling microscopy and spectroscopy, we observed the emergence of a $(2 \times 2)$ charge density wave order in single-layer TiTe$_2$ with a transition temperature of $92 \pm 3$ K. Also observed was a pseudogap of about 28 meV at the Fermi level at 4.2 K. Surprisingly, no charge density wave transitions were observed in two-layer and multi-layer TiTe$_2$, despite the quasi-two-dimensional nature of the material in the bulk. The unique charge density wave phenomenon in the single layer raises intriguing questions that challenge the prevailing thinking about the mechanisms of charge density wave formation.

[1] Department of Physics, University of Illinois at Urbana-Champaign, 1110 West Green Street, Urbana, IL 61801-3080, USA. [2] Frederick Seitz Materials Research Laboratory, University of Illinois at Urbana-Champaign, 104 South Goodwin Avenue, Urbana, IL 61801-2902, USA. [3] Advanced Light Source, Lawrence Berkeley National Laboratory, Berkeley, CA 94720, USA. [4] Center for Condensed Matter Sciences, National Taiwan University, Taipei 10617, Taiwan. [5] Department of Physics, National Taiwan University, Taipei 10617, Taiwan. [6] Institute of Atomic and Molecular Sciences, Academia Sinica, Taipei 10617, Taiwan. [7] Department of Physics, University of Tokyo, Tokyo 113-0033, Japan. [8] School of Physics, Georgia Institute of Technology, Atlanta, GA 30332, USA. Correspondence and requests for materials should be addressed to P.C. (email: pchen229@illinois.edu) or to T.-C.C. (email: tcchiang@illinois.edu)

Ultrathin films have found extensive utility in electronics, optics, and materials research[1–4]. Systems that undergo phase transitions under varying physical parameters such as the temperature or external fields are especially interesting because the response of the system can be exploited for sensor, memory, and logic device applications[5, 6]. Continued demand for device miniaturization has led to the current focus on films that are just one atomic or molecular layer in thickness. At this single-layer limit, the material's physical properties can be fundamentally different from those of the bulk counterpart, and emergence of novel properties is a prevailing research theme[7–9].

A case of interest is TiTe$_2$, which belongs to an extensive family of layered transition metal dichalcogenides MX$_2$, and whose quasi-two-dimensional (2D) structural and electronic properties have been intensively studied during the past few decades[10–15]. Many of the MX$_2$ materials exhibit charge density wave (CDW) transitions, but this is not the case for bulk TiTe$_2$[12, 14, 15]. An interesting contrasting case is the related material TiSe$_2$, which exhibits a bulk (2 × 2 × 2) CDW transition at 205 K[16]. Bulk TiSe$_2$ is an indirect semiconductor with a tiny gap separating the conduction and valence bands[17]. While there is no relevant Fermi surface nesting, the tiny indirect gap can mediate a band-type Jahn-Teller interaction or an excitonic interaction that could drive a CDW transition[17]. By contrast, the bonding in TiTe$_2$ is less ionic than TiSe$_2$. The gap should be substantially different, and in fact the material is a metal/semimetal with a negative band gap of about −0.8 eV[10–12]. The resulting metallic Fermi surfaces, however, do not have regions suitable for nesting. Thus, no CDW is expected nor observed in TiTe$_2$ in accordance with the traditional picture. Both TiTe$_2$ and TiSe$_2$ are made of layers stacked loosely together by van der Waals forces, and no dramatic electronic effects are expected in going from 3D to 2D. Indeed, single-layer TiSe$_2$ shows a CDW transition at just ~ 27 K above the bulk transition temperature[7, 18].

It comes as a surprise that our study based on angle-resolved photoemission spectroscopy (ARPES) and scanning tunneling microscopy and spectroscopy (STM/STS) shows that single-layer TiTe$_2$ exhibits a (2 × 2) CDW transition, but two-layer and multi-layer TiTe$_2$ show no related transitions. Single-layer TiTe$_2$ appears to exemplify the emergence of new physics in the 2D limit. The anomalous behavior of the single layer of TiTe$_2$ calls into question the larger issue of CDW mechanisms in general.

## Results

**Film growth, structure and electron diffraction patterns**. The crystal structure of TiTe$_2$ is illustrated in Fig. 1a. Each Te-Ti-Te trilayer (TL) consists of a triangular Ti atomic layer sandwiched between two triangular Te atomic layers, and the TLs are vertically stacked by van der Waals bonding with a larger inter-TL spacing[10]. The projection of the unit cell onto the TL plane is shown in Fig. 1b. Thin films of TiTe$_2$ were grown in situ on a bilayer-graphene-terminated 6H-SiC (0001)[9, 19]. Empirically, this epitaxial graphene is an excellent substrate for growth of many layered MX$_2$ materials; the weak van der Waals bonding at the interface results in the formation of TiTe$_2$ films that are orientationally aligned with the substrate in-plane crystallographic directions. Reflection high-energy electron diffraction (Fig. 1c) from a 1-TL TiTe$_2$ film reveals a high quality and well-ordered film with an in-plane lattice constant of 3.78 ± 0.04 Å. This value agrees with the bulk value of 3.777 Å, but is incommensurate with the graphene lattice constant of 2.46 Å; the film is thus unstrained by the substrate lattice.

**ARPES spectra and electronic band structure**. Calculated band structures based on the GGA method for 1-TL and bulk TiTe$_2$ in the normal phase (Fig. 1d) are fairly similar. The bulk case appears more complicated because of the additional dispersion in the layer-stacking direction. For 1-TL specifically, the top of the valence bands, dominated by the Ti 3$d$ states, is located at the zone center $\overline{\Gamma}$. The bottom of the conduction band, also dominated by the Ti 3$d$ states, is located at the zone boundary $\overline{M}$. The two band edges are separated by a negative gap of about 0.5 eV, and the system is a semimetal. Bands of a 1-TL sample as observed by ARPES (Fig. 1e) at 150 and 10 K compare well with the calculated 1-TL band structure. The results are generally consistent with prior studies where available[10–12], and all calculated band features are observed in the ARPES map. The 10 K data are sharper due to a reduced thermal broadening. Importantly, the 10 K data show additionally a weak replica of the valence bands shifted from $\overline{\Gamma}$ to $\overline{M}$. The replica bands overlap with the intense conduction band

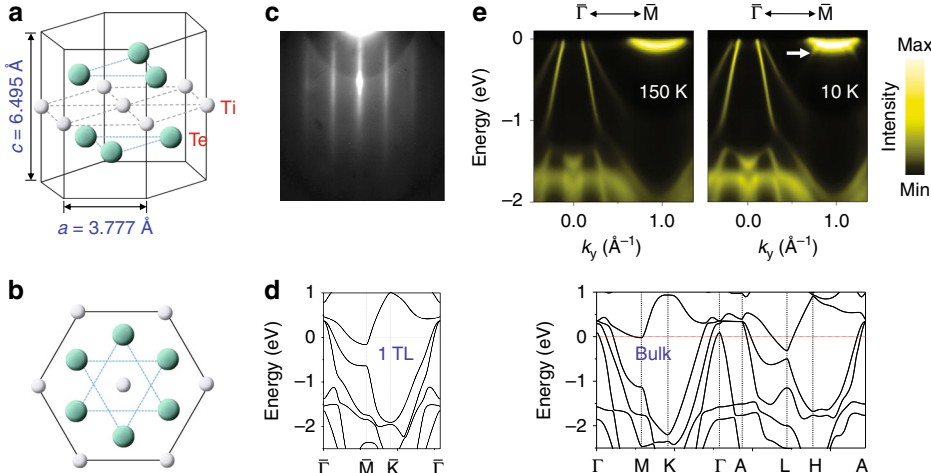

**Fig. 1** Crystal structure and electronic band structure of single-layer TiTe$_2$. **a** Atomic structure of a single layer of TiTe$_2$. The quantities $a$ and $c$ are the lattice constants of bulk TiTe$_2$ taken from ref. [13]. **b** Same structure projected onto the (0001) plane. **c** A RHEED pattern taken after film growth. The spacings between streaks yield a measure of the lattice constant. **d** Calculated band structure for a 1-TL and bulk TiTe$_2$ using the GGA method. **e** ARPES spectra taken from a 1-TL TiTe$_2$ along the $\overline{\Gamma}$-$\overline{M}$ direction for the normal phase at 150 K and the (2 × 2) CDW phase at 10 K. The *arrow* points at features correspond to (2 × 2) folded valence bands

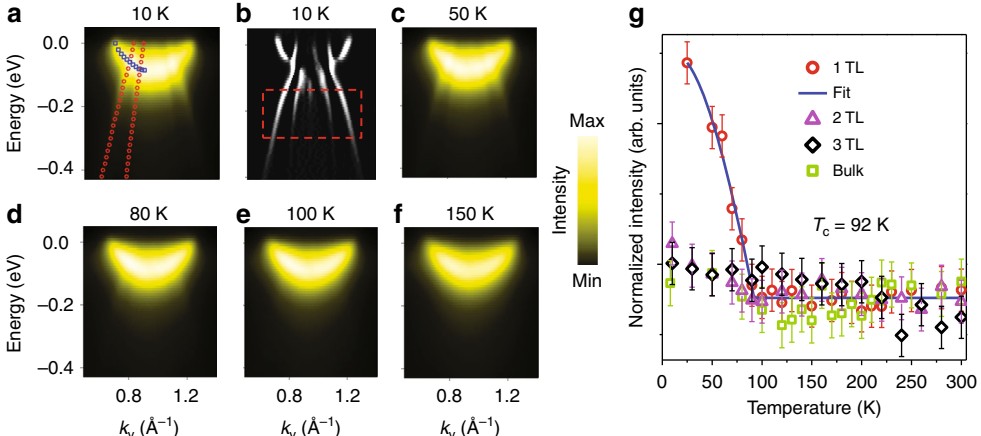

**Fig. 2** Temperature dependence of the folded valence bands and the CDW transition temperature. **a** ARPES map around $\overline{M}$ at the zone boundary taken at 10 K. The *red* and *blue* curves indicate the $(2 \times 2)$ folded valence bands and the conduction band, respectively, on the *left side* only, assuming that the bands do not interact. **b** Second-derivative map of the same data, which highlights the band dispersions. The *red dashed box* indicates a region of interest used for integrating the ARPES intensity as a measure of the folded-band intensity. **c–f** ARPES maps taken at 50, 80, 100, and 150 K, respectively. The folded valence bands diminish as $T$ increases. **g** Integrated ARPES intensities over the region of interest as a function of temperature for the 1-TL, 2-TL, and 3-TL and bulk TiTe$_2$ samples. The *error bar* is deduced from the s.d. of the fitting. The blue curve is a fit using a mean-field equation described in the text

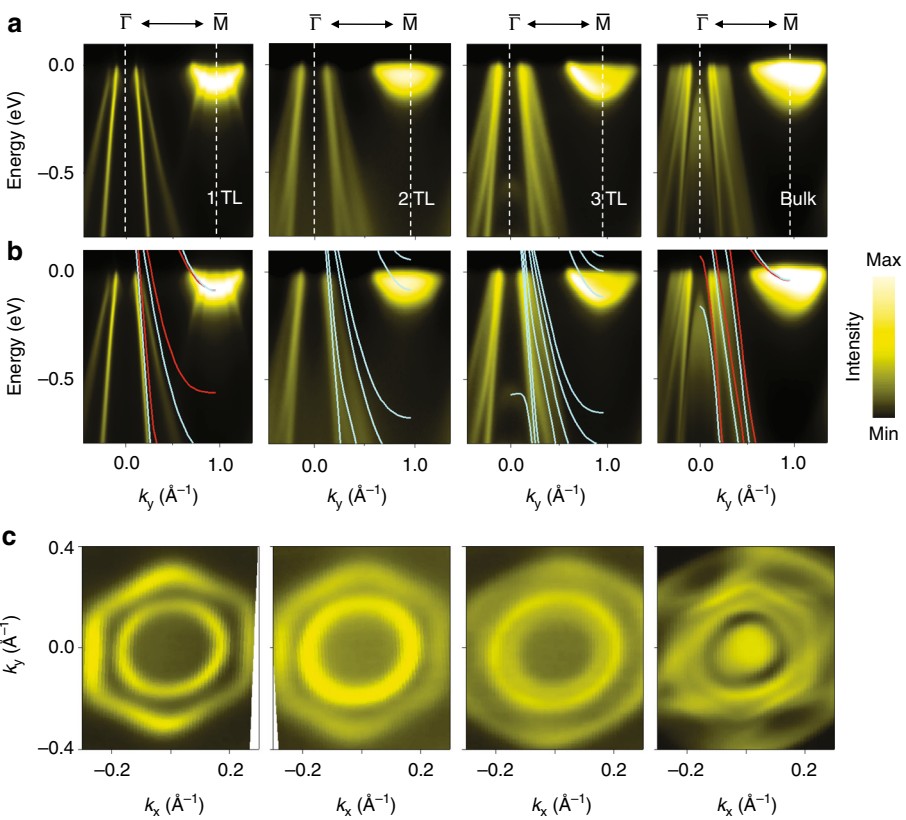

**Fig. 3** ARPES maps of thin-film and bulk TiTe$_2$. **a** ARPES maps, taken at 10 K, for 1-TL, 2-TL, and 3-TL and bulk TiTe$_2$ along the $\overline{\Gamma}$-$\overline{M}$ direction. **b** Same data but superimposed with band structures calculated using GGA (*red curves*) and GGA + *U* (*cyan curves*) for comparison. **c** ARPES constant-energy-contour maps around $\overline{\Gamma}$ at an energy of –0.25 eV

and are most obvious just below the conduction band, as indicated by an arrow. This band replication suggests a $(2 \times 2)$ folding and CDW superlattice formation at 10 K.

**Temperature dependence of the folded bands and CDW transition temperature.** A detailed view of the folded bands around $\overline{M}$ is shown in Fig. 2a, where the blue and red curves

indicate the dispersion relations of the conduction band and the folded valence bands, respectively, assuming that they are not interacting. Second derivatives of this ARPES map (Fig. 2b) reveal additional details. At higher temperatures (Figs. 2c–f), the intensity of the replica bands diminishes and becomes undetectable at 100 K and above. A region of interest, defined by the red dashed rectangle shown in Fig. 2b, is used for ARPES intensity integration as a quantitative measure of the folding

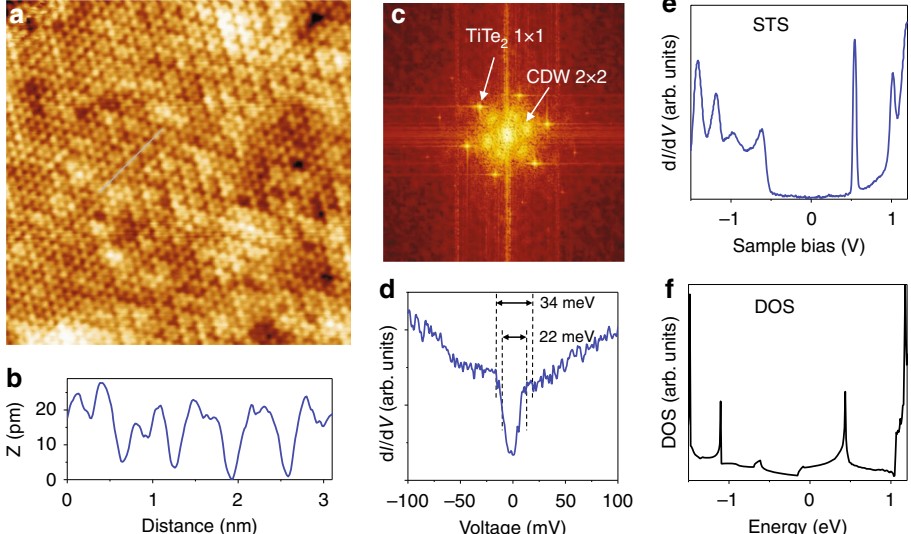

**Fig. 4** STM/STS data for single-layer TiTe$_2$ at 4.2 K. **a** An image taken from 1-TL TiTe$_2$. It shows the triangular lattice of the top Te atomic layer. The experimental conditions are: size 12 × 12 nm, sample bias 105 mV, and tunneling current 1.0 nA. The pattern also shows a weak (2 × 2) modulation that is interrupted by domain boundaries. **b** The height profile along the light blue line, reveals a (2 × 2) height modulation of just about 5 pm. **c** A pattern derived from the Fourier transform of the image. The (1 × 1) lattice spots are sharp; the (2 × 2) CDW spots are weak and broad. **d** A STS differential conductance curve revealing a pseudogap at the Fermi level. **e** STS data over a wide energy range. **f** Computed density of states (DOS) for 1-TL TiTe$_2$. The peak positions are in close agreement with those in the STS data

effect. The integrated intensity as a function of temperature (red circles in Fig. 2g) follows a functional form indicative of a second-order phase transition well described by the mean field behavior

$$I(T) \propto \tanh^2\left(A\sqrt{\frac{T_C}{T} - 1}\right)\Theta(T_C - T) \qquad (1)$$

where $A$ is a proportional constant and $\Theta$ is the unit step function. The blue curve in Fig. 2g is a fit based on this equation. The fit yields $A = 1.37$ and a transition temperature of $T_C = 92 \pm 3$ K.

**Thickness dependence of the band structure.** Bulk TiTe$_2$, on the other hand, does not undergo a CDW transition. An interesting question is how the CDW evolves as additional layers are built up from the single layer toward the bulk limit. As additional TLs are grown over the initial TL, the band structure shows layer-by-layer evolution. The ARPES maps in Fig. 3a for 1-, 2-, and 3-TL and a cleaved bulk TiTe$_2$ were all obtained at 10 K. The calculated band structures based on GGA (red curves) and GGA + $U$ (cyan curves) are included in Fig. 3b for comparison with the data; a Hubbard $U$[20] of 3 eV is included to improve the overall agreement between the calculated and experimental dispersions between 0 and –2 eV. A clear illustration of the band evolution is given by the band contours at –0.25 eV (Fig. 3c). For 1-TL, the contours consist of an outer hexagon and an inner circle, which correspond to the two top valence bands discussed earlier in connection with Fig. 1. For the 2-TL sample, the inner circle is split into two barely resolved circles because of band doubling. The outer hexagon should also split, but the splitting is too small and therefore unresolved. For the 3-TL sample, there should be three inner circles; only two are observed, which are, however, much farther apart than the 2-TL case. Based on a comparison with the theory, the outer circle actually consists of two unresolved circles. The ARPES contour map for the bulk sample should reveal band continuums due to band dispersion along the layer-stacking direction, which can be distorted or modulated by

cross section variations to form a complex pattern as seen in experiment. The integrated ARPES intensity within the region of interest (box in Fig. 2b) as a function of temperature is shown in Fig. 2g for the different samples. Only the 1-TL sample shows a clear CDW transition within experimental error.

**STM/STS evidence for CDW order and a pseudogap.** Real-space evidence for a (2 × 2) CDW in the single layer is provided by STM measurements at 4.2 K. The triangular lattice associated with the top Te atoms is readily observed (Fig. 4a). Period doubling is evident along the three close-packed lattice directions, giving rise to an overall (2 × 2) structure, which is, however, interrupted by domain boundaries. A line profile (Fig. 4b) illustrates the period doubling; the (2 × 2) height modulation, about 5 pm, is very weak. A Fourier transform of the STM image (Fig. 4c) shows sharp (1 × 1) lattice peaks and relatively weak and broad (2 × 2) spots. The broadening can be related to the domain boundaries. A STS differential conductance curve (Fig. 4d), obtained with a 2-mV bias modulation, reveals a 60% conductance dip at the Fermi level. It is not a true gap, but a "pseudogap", because the conductance does not drop to zero. The pseudogap size, about 28 meV, equals 3.5 $k_B T_C$, where $T_C = 92$ K. This numerical relation between the gap and the transition temperature happens to agree with what governs the threshold for a weakly coupled dirty-limit superconductor. STS data taken with the same tip over a wide energy range (Fig. 4e) show a number of peaks that agree closely with those in the computed density of states of a 1-TL TiTe$_2$ (Fig. 4f). This agreement indicates that the pseudogap is a real feature, not an artifact arising from tip effects. The pseudogap was repeatedly observed with calibrated tips at different spots on the sample surface, but the conductance dip is reduced in regions where the (2 × 2) modulation is weaker.

**Discussion**

The measured and calculated band structures of single-layer TiTe$_2$ do not show any nesting at the (2 × 2) period (Supplementary Notes 2 and 3). The excitonic mechanism, a theory often invoked to explain CDWs in systems that do not exhibit Fermi

surface nesting, presumes a small gap; it does not apply to the semimetallic single-layer TiTe$_2$; in any case, any excitonic interactions would get screened out by the abundant charge carriers in this semimetal. A band-type Jahn-Teller interaction, another theory for CDW transitions, would give rise to a repulsion between the conduction and valence bands, thus lowering the total valence electronic energy. The present case does not fit neatly into this model. However, band folding (Fig. 2b) should give rise to gap opening at the intersection points between the conduction and folded valence bands. Ideally, if the gap opening happened at the Fermi level, the valence states nearby would get pushed to lower energies, thus reducing the total electronic energy. In reality, the intersection points around the Fermi level happen over a range of energy around the Fermi level (Supplementary Fig. 2). It is possible that a net reduction of system energy still happens, thus favoring the (2 × 2) structure.

To explore further if this mechanism in terms of band structure modification would work, we have performed calculations in which an assumed (2 × 2) structure of TiTe$_2$ is allowed to relax toward its energy minimum. However, the outcome is inconclusive, since the energy difference between the (1 × 1) structure and a slightly distorted (2 × 2) structure is within the numerical accuracy in the calculation. Calculated phonon dispersion relations for bulk and single-layer TiTe$_2$ (Supplementary Note 7) show no imaginary frequencies that would correspond to structural instabilities. The small dip in an acoustic branch at $\overline{\mathrm{M}}$ for the single layer might suggest an enhanced electron-phonon coupling compared to the bulk and thus a tendency for (2 × 2) distortion. The results from these calculations do agree with the experiment on one issue; that is, the (2 × 2) CDW phase must be a very weak effect, if it happens. Interaction of the film by van der Waals bonding with the substrate, although weak and not included in the calculation, is another factor to consider. The interaction, being incommensurate, gives rise to essentially random perturbations with no coherent consequences; the net effect is like scattering, which should suppress the tendency for ordering. Also, the measured ARPES band structure corresponds closely to theoretical results for a freestanding film. These considerations suggest that interaction with the substrate is not a key factor for the formation of the CDW phase.

While band structure modification leading to energy lowering appears to be the most likely explanation for the (2 × 2) CDW, two issues remain mysterious. One is that 2- and multi-layer TiTe$_2$ show no related CDW transitions. Of course, it is possible that adding the dispersion in the layer-stacking direction somehow suppresses the energy lowering effect, but again we have not been able to show this numerically based on band calculations. It also goes against the seemingly related case of TiSe$_2$ where the bulk and single layer have similar transition temperatures, as expected for a quasi-2D system. The other issue is the origin of the pseudogap at the Fermi level. Band folding leads to intersection of the conduction and valence bands. Gaps would open, but these are distributed over an energy range much larger than the STS gap size (Supplementary Note 1). Whether there are any additional many-body interactions that would lead to the observed pseudogap is an interesting theoretical question, and its resolution could have strong implications regarding the physics of CDWs in general. TiTe$_2$ appears to be a unique case where the 2D limit is a singular point with unexpected physical properties that are yet to be explained in full.

## Methods

**Experimental details.** ARPES measurements were performed at beamlines 12.0.1 and 10.0.1, Advanced Light Sources (ALS), Lawrence Berkeley National Laboratory; each of the beamline was equipped with a connected molecular beam epitaxial growth system. Substrates of 6H-SiC(0001) were flash-annealed for multiple cycles to form a well-ordered bilayer graphene on the surface[19]. Films of TiTe$_2$ were grown on top of the bilayer graphene by co-evaporating Ti and Te from an electron-beam evaporator and a Knudsen effusion cell, respectively, while the substrate was maintained at 300 °C. The growth process was monitored by reflection-high-energy-electron diffraction, and the growth rate was set to 30 min per TL of TiTe$_2$. Completion of a TL followed by formation of each additional TL of TiTe$_2$ was evident from the development of subbands corresponding to different layer thicknesses in the ARPES map; this procedure permitted preparation of films of specified thicknesses. For measurements of bulk TiTe$_2$, a single crystal was purchased from HQ Graphene; it was cleaved in situ in the experimental chamber to expose a fresh surface. The crystallographic orientation of each sample was determined by electron diffraction and, more precisely, from the pattern symmetry of constant-energy ARPES contours. Some of the film samples were capped with a 20-nm protective layer of Te and then transferred to a separate STM/STS system, wherein the protective Te layer was thermally desorbed at 250 °C before measurements using an Omicron LTSTM instrument and freshly flashed tungsten tips.

**Computational details.** We performed the first-principles calculations using the Vienna ab initio package (VASP)[21–23] with the projector augmented wave method[24, 25]. A plane wave energy cutoff of 320 eV and an $18 \times 18 \times 12$ ($24 \times 24 \times 1$) $k$-mesh were employed for the bulk (1-TL) system. The generalized gradient approximation (GGA) with the Perdew-Burke-Ernzerhof (PBE) functional[26] was used for structural optimization of multiple-layer TiTe$_2$. Freestanding films are modeled with a 15-Å vacuum gap between adjacent layers in the supercell. The PBE functional is known to overestimate the interlayer distance for multilayers of van der Waals solids. Therefore, we applied the van der Waals correction with the DFT-D3 method[27] in multiple-layer calculations in order to optimize the interlayer distance. The in-plane lattice constants are also optimized independently. The resulting in-plane lattice constants (interlayer distances) are 3.76, 3.74 (6.61), 3.74 (6.60), and 3.74 (6.58) Å for single, double, triple, and quadruple layers, respectively. Overall, the results are close to the experimental lattice constants of 3.777 (6.495) Å for the bulk. We performed phonon calculations using the supercell method as implemented in the Phonopy package[28]. A higher energy cutoff of 550 eV and a denser $28 \times 28 \times 1$ $k$-mesh were used to obtain converged phonon results.

**Data availability.** The data that support the findings of this study are available within the article or from the corresponding author upon request.

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

## Acknowledgements

This work is supported by the US Department of Energy, Office of Science, Office of Basic Energy Sciences, Division of Materials Science and Engineering, under Grant No. DE-FG02-07ER46383 (TCC), the National Science Foundation under Grant No. EFMA-1542747 (MYC), the Ministry of Science and Technology of Taiwan under Grant No. 103-2923-M-002-003-MY3 (WWP), and JSPS KAKENHI Grant Nos. JP16 H02108, JP25110010, and JP15K17464 (AT and SH). The Advanced Light Source is supported by the Director, Office of Science, Office of Basic Energy Sciences, of the US Department of Energy under Contract No. DE-AC02-05CH11231. The work at Academia Sinica is supported by a Thematic Project.

## Author contributions

P.C. and T.C.C. designed the project. P.C. with the aid of C.Z.X., S.K.M., A.V.F., and T.C.C. performed MBE growth, ARPES measurements, and data analysis. Y.H.C. and M.Y.C. performed first-principles calculations. W.W.P. conducted STM/STS experiments, and A.K. helped with related data analysis. A.T and S.H. performed transport measurements. T.C.C., P.C., W.W.P., and M.Y.C. interpreted the data. T.C.C. and A.V.F. jointly led the ARPES project. P.C. and T.C.C. wrote the paper with input from other coauthors.

## Additional information

**Competing interests:** The authors declare no competing financial interests.

