## [Peer Review File · Nature Communications]

Reviewers' Comments:

Reviewer #1:

Remarks to the Author:

The paper is interesting: the authors find that single-layered TiTe₂ exhibits a CDW state whereas the bulk does not and reasons for this are discussed. The paper is generally well-written and the ARPES measurements are particularly nice.

However, I believe there a number of questions which the authors should address before the manuscript is considered/accepted for publication.

1. The authors mentioned that van der Waals forces between the substrate and the TiTe₂ film ensures alignment of crystal axes but the films are "otherwise unstrained." I think this statement needs to be further supported.

The authors do point to RHEED measurements indicating independent lattice constants. What are these lattice constants? How do lattice parameters of the TiTe₂ film compare to that of bulk TiTe₂? Such a comparison would be helpful in ensuring that the substrate is not straining the TiTe₂ 1-TL lattice.

Along these lines, for the lattice parameters in Figure 1a: presumably the c-axis is based on bulk lattice parameters. Is the a-axis from the RHEED measurements in Figure 1b or taken from elsewhere? A value of $a = 3.777$ angstroms is the bulk value at least reported by Arnaud and Chevreton, J. Solid State Chem. 39, 230, 2981. The lattice parameters from RHEED should be clearly stated and compared to this bulk value.

2. I think the authors should address the possible role of the substrate on their measurements. Van der Waals forces hold the layers in bulk TiTe₂ together and such forces are present between the single-layer TiTe₂ and the substrate. Why/how can we thus ignore the substrate here?

Along these lines, in a previous study of single-layer TiSe₂ (Chen et al., Nature Communications, 6, 8943, 2015 - which includes several of the same authors as on the current manuscript), the authors note that a higher CDW T_{cdw} in single-layer than the bulk indicating that the "CDW phase in the single layer is more robust." And, in the same paragraph they write "The random interface potential caused by the lattice mismatch between graphene and TiSe₂ could suppress the CDW T_c relative to the other cases." In their previous paper, they addressed the possible role of the substrate, why not here? I think this is an important issue to address.

3. The authors mention a 2x2 superstructure "interrupted by domain boundaries." The FFT of the topographic image provides convincing evidence of 2x2 superstructure which can also be seen by eye in many places in the topography. A few questions along these lines:

a. Is the 2x2 superstructure everywhere across the surface? At many places it looks to be absent. If it is absent, does this mean there are CDW fluctuations which are being pinned leading to a static structure? Or, are there reasons why the authors can suggest why the CDW may be locally suppressed? Is this further evidence of the CDW phase being weak in TiTe₂ (and then can the authors use STM on the atomic scale to see what causes the CDW to be disrupted)?

b. What drives the inhomogeneity (structural and electronic) seen in the image? Are these driven by variations in the 6H-SiC (0001) substrate (perhaps indicating the importance of the substrate on what is happening within the TiTe₂ film) or imperfections in the crystal lattice of the TiTe₂ film?

c. What drives the domains? Can the edges of a domain be traced on the figure to clearly illustrate

to the readers such a domain region? Perhaps Fourier filtering just the 2x2 lattice in the topography would clearly identify domain regions.

4. The pseudogap found in STM is interesting as well as the several features seen at higher energies in Figure 4d. The authors also indicate that this pseudogap is seen with multiple tips at different locations on the surface sample and so is a real feature. Is this gap seen only in regions where the 2x2 superstructure is most obvious/present? The authors note that equating this gap size to 3.5 kT gives a transition of 92 K, very similar to extracted transition temperature for the CDW based on ARPES measurements. Can this gap be seen in ARPES measurements?

5. In discussing the mechanism for the CDW in TiTe₂, I find particularly lacking the discussion of the possibility of the importance of electron-phonon coupling which is often reported as important in understanding CDW states in related transition metal dichalcogenides. Further, such CDW enhancement within a single-layer film over the bulk has been seen in other transition metal dichalcogenides such as NbSe₂ where the CDW transition temperature goes from 33 K (in the bulk) to 145 for a single layer. (Xi et al., Nature Nanotechnology, 10, 765, 2015) And, in this study of single-layer NbSe₂, the authors find "the increasing electron-phonon coupling strength is probably the major driving mechanism for stronger CDWs in atomically thin NbSe₂."

Could it be the same in TiTe₂? Why or why not?

Is it possible that what the authors see in TiTe₂ is an extension of what happens in NbSe₂? For example, in NbSe₂, the CDW is weak in the multilayer and enhanced in the thin film due to enhanced el-ph coupling. In the TiTe₂ bulk, the CDW is so weak that it cannot form, but, in a single layer it can emerge due to similar enhancement of el-ph coupling? And, because the CDW is weaker in bulk TiTe₂ than in NbSe₂ this leads to a T_{cdw} which is lower in single-layer TiTe₂ than in single-layer NbSe₂?

Reviewer #2:

Remarks to the Author:

The manuscript reports on the study of thin films of TiTe₂ which belongs to the family of transition metal dichalcogenides with quasi two dimensional structures exhibiting a rich variety of different phase transitions. The ARPES and STM/STS measurements in a wide temperature range together with first-principles calculations are used in the present work to reveal any evidences of the charge density wave which may appear in TiTe₂.

In recent decades, several papers have been published which contain results indicating the possible formation of a charge density wave in bulk samples TiTe₂: an anomaly on the temperature dependence of the magnetic susceptibility observed on a bulk sample [12]; unusual non-linear I-V characteristics measured on the exfoliated TiTe₂ film [doi:10.1063/1.3679679]. There is also evidence that CDW seemingly appears in the intercalated compounds based on TiTe₂, in particular, in CrxTiTe₂ [doi:10.1088/0953-8984/21/50/506002]. However, up to now there has been no unequivocal answer to the question whether a charge density wave in TiTe₂ is formed analogous to that observed in the isostructural TiSe₂ compound.

This work contains new and very interesting data on the temperature evolution of the electronic structure of the TiTe₂ films of different thickness; and the period doubling (2x2) is derived from the STM measurements on a single-layer film. These results definitely add physics of transition metal dichalcogenides and CDW transitions.

However, in my opinion, additional information on the behavior of the electrical resistivity in TiTe₂ films of different thicknesses is undoubtedly necessary for the publication of this article in the Nature Communications. This is because the formation of a CDW is always accompanied by a change in the electrical resistance, since the anomalous behavior of the electrical resistivity is a

simple and characteristic indicator of the transition to the CDW state. This is shown for all known materials experiencing the transition to the CDW state. Moreover, the above mentioned literature data indicating the possibility of the CDW state in TiTe₂ should be briefly discussed.

Reviewer #3:

Remarks to the Author:

The authors have studied the electronic properties of a 2D material, TiTe₂, for various thickness ranging from bulk to monolayer limit. Using the ARPES, STM and STS techniques, the authors have reported that monolayer of TiTe₂ undergoes a charge density wave (CDW) phase transition below the temperature of about 92 K and subsequently it forms a 2x2 superstructure. Interestingly the CDW phase disappears in any other thicker layers of TiTe₂, larger than the monolayers one. The results look interesting. Note that previously the authors had been reported the identical experiments for TiSe₂ (iso-electronic to TiTe₂) and published in Nature communication (Nat. Commun. 2015, reference 7). Therefore I did not find any additional technological advancement/challenges for the current experiments set up. However, the significant achievement in their present manuscript is the finding of the CDW phase only in monolayer of TiTe₂ not in bulk, where as it exists in both bulk and monolayer of TiSe₂. Such a novelty (with some further clarifications) may be considered for publishing in Nature Communication article. The appearance of the CDW phase can be ascribed to several competing mechanisms such as Peierls instability, exciton insulator instability, Jahn-Teller distortion, or Fermi surface nesting. Although the authors addresses some of the mechanism using theoretical calculations, still I feel that their discussions are confusing and needed to be addressed carefully. Therefore the authors need to address the issues listed below before considering the manuscript to be published in Nature Communication.

- 1) For TiSe₂, superconductivity and CDW phase do coexist and the phenomenon are likely to be driven due to presence of strong electron-phonon coupling (PRL, 106, 196406 (2011)) in these kind of layered chalcogenides. Such a strong electron-phonon coupling may drive to the aforementioned coexistence in monolayer of TiTe₂ and this kind of clarification will give more insights about the phase diagram and the underline mechanism of such phase transition.
- 2) The authors mentioned the spin-orbit splitting of the TiTe₂ band structure, although the system has inversion symmetry. Therefore it will be interesting if the authors discuss the mechanism behind the spin-orbit splitting.
- 3) The authors mentioned that graphene and TiTe₂ are connected by weak vdW forces. Moreover, TiTe₂ layers are itself weakly bonded by vdW forces and the electronic structure does not change significantly due to dimensional reduction from the bulk to monolayer limit. However, the CDW phase is only present in the monolayer limit. Therefore role of substrate is needed to be discussed more in details. Is there are any effects of doping or strain from the substrate?
- 4) As the monolayer of TiTe₂ exhibits the CDW phase, multilayer of TiTe₂ likely to have Kohn anomaly in their phonon band-structure, whereas unstable phonon mode appears in the monolayer case (EPL 115, 47001 (2016)).
- 5) As the authors reported the formation of 2x2 super-structure for monolayer TiTe₂ in CDW phase, it would be interesting to show/report the strength of atomic displacements.
- 6) As the DFT-semi-local functional usually underestimate the bandgap, sometimes it is not promising in simulating the ARPES spectral weight accurately. In fact the their DFT calculation shows that 2x2 distorted super cell is not energetically favorable in the CDW phase. Such an ambiguity is needed to be addressed with more accurate calculations in their DFT calculations. Moreover the electronic band-structure would largely depend on the experimental growth

environments, which needed to be discussed.

7) The authors have mentioned that due to metallic/semi-metallic behavior, exciton mechanism can be discarded. The exciton effect may have significant effect for such a monolayer thickness in which the confinement plays a crucial role. Previous study shows that exciton effect is significant even in semi-metallic graphene (Nature Nanotechnology 5, 32 (2009)). As the energetic of the electronic structure is inconclusive, the role of exciton may provide some useful insights.

8) Although the authors have mentioned that distorted super-cell is not energetically favorable, it is very confusing to plot the DOS (Fig.S4) at different lattice constant. If the authors wanted to simulate the pressure effect, they need to discuss the effect in more details by discussing both the phonon and electronic structure.

Comments of Reviewer #1: The paper is interesting: the authors find that single-layered TiTe₂ exhibits a CDW state whereas the bulk does not and reasons for this are discussed. The paper is generally well-written and the ARPES measurements are particularly nice.

However, I believe there a number of questions which the authors should address before the manuscript is considered/accepted for publication.

Authors' response: We thank the reviewer for finding our results nice and interesting. We address the reviewer's concerns, point-by-point, in the following.

1. The authors mentioned that van der Waals forces between the substrate and the TiTe₂ film ensures alignment of crystal axes but the films are "otherwise unstrained." I think this statement needs to be further supported.

The authors do point to RHEED measurements indicating independent lattice constants. What are these lattice constants? How do lattice parameters of the TiTe₂ film compare to that of bulk TiTe₂? Such a comparison would be helpful in ensuring that the substrate is not straining the TiTe₂ 1-TL lattice.

Along these lines, for the lattice parameters in Figure 1a: presumably the c-axis is based on bulk lattice parameters. Is the a-axis from the RHEED measurements in Figure 1b or taken from elsewhere? A value of $a = 3.777$ angstroms is the bulk value at least reported by Arnaud and Chevreton, J. Solid State Chem. 39, 230, 2981. The lattice parameters from RHEED should be clearly stated and compared to this bulk value.

Authors' response: We thank the reviewer for pointing out the details. The lattice parameters in Fig. 1a are the bulk values taken from Ref. 13; this is now clarified in the caption of the figure. The in-plane lattice constant of single layer TiTe₂ from our RHEED measurements is 3.78 ± 0.04 Å. This value agrees with the bulk value of 3.777 Å, but is incommensurate with the graphene lattice constant of 2.46 Å; the film is thus unstrained by the substrate lattice. We have rewritten the subsection "Film growth, structure and electron diffraction patterns" within the "Results" section to include the lattice constants and to show explicitly that the film is incommensurate with and unstrained by the substrate lattice.

2. I think the authors should address the possible role of the substrate on their measurements. Van der Waals forces hold the layers in bulk TiTe₂ together and such forces are present between the single-layer TiTe₂ and the substrate. Why/how can we thus ignore the substrate here?

Along these lines, in a previous study of single-layer TiSe₂ (Chen et al., Nature Communications, 6, 8943, 2015 - which includes several of the same authors as on the current manuscript), the authors note that a higher CDW T_{cdw} in single-layer than the bulk indicating that the "CDW phase in the single layer is more robust." And, in the same paragraph they write "The random interface potential caused by the lattice mismatch between graphene and TiSe₂ could suppress the CDW T_c relative to the other cases." In their previous paper, they addressed the possible role of the substrate, why not here? I think this is an important issue to address.

Authors' response: We agree with the referee's suggestion to include a discussion of possible effects of the substrate. The referee is quite correct that we discussed this issue in our previous study of TiSe_2 , and a similar discussion would be desirable here. We have added the following sentences to the discussion:

Interaction of the film by van der Waals bonding with the substrate, although weak and not included in the calculation, is another factor to consider. The interaction, being incommensurate, gives rise to essentially random perturbations with no coherent consequences; the net effect is like scattering, which should suppress the tendency for ordering. Also, the measured ARPES band structure corresponds closely to theoretical results for a freestanding film. These considerations suggest that interaction with the substrate is not a key factor for the formation of the CDW phase.

3. The authors mention a 2×2 superstructure “interrupted by domain boundaries.” The FFT of the topographic image provides convincing evidence of 2×2 superstructure which can also be seen by eye in many places in the topography. A few questions along these lines:

a. Is the 2×2 superstructure everywhere across the surface? At many places it looks to be absent. If it is absent, does this mean there are CDW fluctuations which are being pinned leading to a static structure? Or, are there reasons why the authors can suggest why the CDW may be locally suppressed? Is this further evidence of the CDW phase being weak in TiTe_2 (and then can the authors use STM on the atomic scale to see what causes the CDW to be disrupted)?

Authors' response: The CDW superlattice distortion is extremely weak in single layer TiTe_2 . The domain sizes are small and fairly irregular. As a result, the 2×2 structure cannot be clearly identified over a portion of the surface near the domain boundaries and junctures. There are several possibilities for the domain structures. For example, impurities and defects in the substrate at the surface and below the surface could create local perturbations and pin or disrupt the CDW superlattice structure. Defects in the film can be another source for the partial disorder. This discussion is now included in Section 6 of the supplementary document.

b. What drives the inhomogeneity (structural and electronic) seen in the image? Are these driven by variations in the 6H-SiC (0001) substrate (perhaps indicating the importance of the substrate on what is happening within the TiTe_2 film) or imperfections in the crystal lattice of the TiTe_2 film?

Authors' response: This is explained in our response to question 3a above.

c. What drives the domains? Can the edges of a domain be traced on the figure to clearly illustrate to the readers such a domain region? Perhaps Fourier filtering just the 2×2 lattice in the topography would clearly identify domain regions.

Authors' response: As discussed above, defects and impurities in the film and in the substrate at various depths can create random perturbations. With a (2×2) superstructure, random pinning can give rise to antiphase domains; the domain boundaries become topological defects. Specifically, an antiphase domain is formed if a portion of the original lattice is shifted by one unit vector.

This type of defects is common in systems with multiple atoms in a unit cell. We have tried Fourier filtering as suggested by the referee, but the results are not particularly illuminating for delineating the domain boundaries with irregular shapes. For a better illustration, we have added a new Figure S6 in the supplementary document, which is also shown below. It is derived from the STM image shown in Fig. 1 in the main manuscript by merging the two derivative images in the X and Y directions, followed by slight Gaussian smoothing. The three sets of colored lines are oriented along symmetry-equivalent close-packed directions. In each set, the two lines are drawn along close packed chains of atoms on the surface in two neighboring domains. The offset corresponds to a (1×1) unit cell translation, or one half of a (2×2) unit cell translation. This half translation gives rise to a domain boundary at the closest point of the two lines. This explanation is now included in the supplementary document.

4. The pseudogap found in STM is interesting as well as the several features seen at higher energies in Figure 4d. The authors also indicate that this pseudogap is seen with multiple tips at different locations on the surface sample and so is a real feature. Is this gap seen only in regions where the 2×2 superstructure is most obvious/present? The authors note that equating this gap size to 3.5 kT gives a transition of 92 K , very similar to extracted transition temperature for the CDW based on ARPES measurements. Can this gap be seen in ARPES measurements?

Authors' response: The CDW pseudogap is readily seen in regions where the (2×2) superstructure is obvious. In regions with weaker (2×2) ordering, the pseudogap tends to be weaker. We have added a clarifying remark in the manuscript. The relevant sentence now reads:

The pseudogap was repeatedly observed with calibrated tips at different spots on the sample surface, but the conductance dip is reduced in regions where the (2×2) modulation is weaker.

Regarding the referee's second question, a gap feature is indeed seen with ARPES, as discussed in detail in Section 1 of the original supplementary document. This section remains unchanged in the revised version.

5. In discussing the mechanism for the CDW in TiTe_2 , I find particularly lacking the discussion of the possibility of the importance of electron-phonon coupling which is often reported as important in understanding CDW states in related transition metal dichalcogenides. Further, such CDW enhancement within a single-layer film over the bulk has been seen in other transition metal dichalcogenides such as NbSe_2 where the CDW transition temperature goes from 33 K (in the bulk) to 145 for a single layer. (Xi et al., Nature Nanotechnology, 10, 765, 2015) And, in this study of single-layer NbSe_2 , the authors find “the increasing electron-phonon coupling strength is probably the major driving mechanism for stronger CDWs in atomically thin NbSe_2 .”

Could it be the same in TiTe_2 ? Why or why not?

Is it possible that what the authors see in TiTe_2 is an extension of what happens in NbSe_2 ? For example, in NbSe_2 , the CDW is weak in the multilayer and enhanced in the thin film due to enhanced el-ph coupling. In the TiTe_2 bulk, the CDW is so weak that it cannot form, but, in a single layer it can emerge due to similar enhancement of el-ph coupling? And, because the CDW is weaker in bulk TiTe_2 than in NbSe_2 this leads to a T_{cdw} which is lower in single-layer TiTe_2 than in single-layer NbSe_2 ?

Authors' response: The referee makes an interesting comparison with NbSe_2 , for which the CDW transition temperature of 33 K in the bulk becomes 145 K in a single layer; the results possibly suggest an enhanced electron-phonon coupling in the single-layer leading to the $\sim 4x$ increase of the transition temperature (Ref. 8). Another case of interest is TiSe_2 , which has an electronic structure resembling that of TiTe_2 (while the electronic structure of NbSe_2 looks very different). TiSe_2 exhibits a CDW transition at 205 K in the bulk and 232 K in a single layer (Ref. 7). The interpretation there is different; each layer forms (2×2) first, and then at a lower temperature the layers phase lock and freeze into the bulk $(2 \times 2 \times 2)$ CDW structure. Both interpretations are possibilities for TiTe_2 , but it is hard to prove one way or the other. As pointed out in the main text, the TiTe_2 case is unusual and mysterious because bulk TiTe_2 , unlike NbSe_2 and TiSe_2 , does not exhibit a CDW transition. In fact, no CDW transition is observed even at just two layers of TiTe_2 . The 92 K transition in single-layer TiTe_2 , accompanied by a pseudogap that cannot be explained by any of the known theories, points to possibly unconventional mechanisms.

Another related issue of interest is that superconductivity in bulk TiTe_2 has not been detected down to a temperature of 0.45 K at ambient pressure (Ref. 14). The lack of a superconducting transition and a CDW transition in the bulk may imply weak electron-phonon coupling effects in this system. To look for further clues, we have computed the phonon dispersion relations for a (1×1) single layer and bulk TiTe_2 , which are shown below. There is nothing unusual about the bulk dispersion relations. The small dip in an acoustic branch at \bar{M} for the single layer might suggest a tendency for (2×2) distortion, but there are no imaginary frequencies that would

correspond to structural instabilities. Within our numerical accuracy, we do not find a (2x2) transition.

The above discussion is now included in the supplementary document (Section 7).

Comments of Reviewer #2: The manuscript reports on the study of thin films of TiTe₂ which belongs to the family of transition metal dichalcogenides with quasi two dimensional structures exhibiting a rich variety of different phase transitions. The ARPES and STM/STS measurements in a wide temperature range together with first-principles calculations are used in the present work to reveal any evidences of the charge density wave which may appear in TiTe₂. In recent decades, several papers have been published which contain results indicating the possible formation of a charge density wave in bulk samples TiTe₂: an anomaly on the temperature dependence of the magnetic susceptibility observed on a bulk sample [12]; unusual non-linear I-V characteristics measured on the exfoliated TiTe₂ film [doi:10.1063/1.3679679]. There is also evidence that CDW seemingly appears in the intercalated compounds based on TiTe₂, in particular, in Cr_xTiTe₂ [doi:10.1088/0953-8984/21/50/506002]. However, up to now there has been no unequivocal answer to the question whether a charge density wave in TiTe₂ is formed analogous to that observed in the isostructural TiSe₂ compound. This work contains new and very interesting data on the temperature evolution of the electronic structure of the TiTe₂ films of different thickness; and the period doubling (2x2) is derived from the STM measurements on a single-layer film. These results definitely add physics of transition metal dichalcogenides and CDW transitions.

Authors' response: We thank the reviewer for finding our results new, interesting, and useful for understanding the physics of transition metal dichalcogenides and CDWs.

However, in my opinion, additional information on the behavior of the electrical resistivity in TiTe₂ films of different thicknesses is undoubtedly necessary for the publication of this article in the Nature Communications. This is because the formation of a CDW is always accompanied by a change in the electrical resistance, since the anomalous behavior of the electrical resistivity is a simple and characteristic indicator of the transition to the CDW state. This is shown for all known materials experiencing the transition to the CDW state.

Moreover, the above mentioned literature data indicating the possibility of the CDW state in TiTe₂ should be briefly discussed.

Authors' response: Although de Boer reported anomalies at 150 K from magnetic susceptibility and resistivity measurements of bulk TiTe₂ (Ref. 12), these unusual features were later attributed to nonstoichiometry of the sample, which contained ~1% excess Ti. These observations were not reproduced in later transport measurements on stoichiometric crystals (Refs. 14 and 15). There have been no other experimental observations of such anomalies since.

The referee suggested transport measurements. He/she is quite correct that transport measurements of CDW transitions in *bulk crystals* are common and helpful, but we are dealing with a single layer, which presents a substantial challenge. As far as we know, nobody has ever detected a CDW using transport in a single layer. Nevertheless, we are pleased to add Section 5 in the supplementary document to show new transport data, collected in collaboration with a leading group in Japan. The measurements were performed under UHV using a micro-four-point probe system. The results are shown below and also in the supplementary document (Section 5). Upon decreasing temperature, the sheet resistance shows a break or an onset at ~90 K, below which the resistance increases substantially. The behavior suggests a gap opening. These results agree with the CDW transition at 92 K as observed by ARPES and the pseudogap revealed by STS and ARPES.

We have added A. Takayama and S. Hasegawa to the author list, who performed the transport measurements. They have agreed to be co-authors of our article.

Comments of Reviewer #3: The authors have studied the electronic properties of a 2D material, TiTe₂, for various thickness ranging from bulk to monolayer limit. Using the ARPES, STM and STS techniques, the authors have reported that monolayer of TiTe₂ undergoes a charge density wave (CDW) phase transition below the temperature of about 92 K and subsequently it forms a

2x2 superstructure. Interestingly the CDW phase disappears in any other thicker layers of TiTe₂, larger than the monolayers one. The results look interesting. Note that previously the authors had been reported the identical experiments for TiSe₂ (iso-electronic to TiTe₂) and published in Nature communication (Nat. Commun. 2015, reference 7). Therefore I did not find any additional technological advancement/challenges for the current experiments set up. However, the significant achievement in their present manuscript is the finding of the CDW phase only in monolayer of TiTe₂ not in bulk, where as it exists in both bulk and monolayer of TiSe₂. Such a novelty (with some further clarifications) may be considered for publishing in Nature Communication article. The appearance of the CDW phase can be ascribed to several competing mechanisms such as Peierls instability, exciton insulator instability, Jahn-Teller distortion, or Fermi surface nesting. Although the authors addresses some of the mechanism using theoretical calculations, still I feel that their discussions are confusing and needed to be addressed carefully. Therefore the authors need to address the issues listed below before considering the manuscript to be published in Nature Communication.

Authors' response: We thank the reviewer for finding our results novel.

1) For TiSe₂, superconductivity and CDW phase do coexist and the phenomenon are likely to be driven due to presence of strong electron-phonon coupling (PRL, 106, 196406 (2011)) in these kind of layered chalcogenides. Such a strong electron-phonon coupling may drive to the aforementioned coexistence in monolayer of TiTe₂ and this kind of clarification will give more insights about the phase diagram and the underline mechanism of such phase transition.

Authors' response: Superconductivity is found in TiSe₂ only upon doping or under pressure. For example, the superconducting phase in TiSe₂ sets in only within the pressure range of 2-4 GPa [PRL 103, 236401 (2009)], and this was explained by enhanced electron-phonon coupling under pressure [PRL 106, 196406 (2011)]. Superconductivity in TiTe₂ has not been detected for temperatures down to 0.45 K at ambient pressure (Ref. 14). These findings indicate that the effects of electron-phonon coupling are not significant in these cases unless the materials are properly doped or under pressure. The temperature-pressure or temperature-doping phase diagram would definitely be an interesting topic for TiTe₂. We thank the referee for the suggestion for further work in the future.

2) The authors mentioned the spin-orbit splitting of the TiTe₂ band structure, although the system has inversion symmetry. Therefore it will be interesting if the authors discuss the mechanism behind the spin-orbit splitting.

Authors' response: The top of the valence bands and the bottom of the conduction bands are both mostly associated with the Ti 3d orbitals. The spin-orbit splitting is extremely small and not visible in the figure. This effect does not play an important role in our discussion of the band structure. We have revised the text to avoid unintended implications.

3) The authors mentioned that graphene and TiTe₂ are connected by weak vdW forces. Moreover, TiTe₂ layers are itself weakly bonded by vdW forces and the electronic structure does not change significantly due to dimensional reduction from the bulk to monolayer limit. However, the CDW phase is only present in the monolayer limit. Therefore role of substrate is

needed to be discussed more in details. Is there are any effects of doping or strain from the substrate?

Authors' response: The referee is quite correct. The coupling between the overlayer and the bilayer graphene is weak not only because the interaction is of the van der Waals type, but also because the interface is incommensurate. An incommensurate interaction gives rise to essentially random perturbations with no coherent consequences; the net effect is like scattering, which should suppress the tendency of ordering. In our work, the measured ARPES band structure corresponds closely to theoretical results for a freestanding film. These considerations suggest that interaction with the substrate is not a key factor. This question is similar to that from referee 1. We have added the following sentences to the discussion:

Interaction of the film by van der Waals bonding with the substrate, although weak and not included in the calculation, is another factor to consider. The interaction, being incommensurate, gives rise to essentially random perturbations with no coherent consequences; the net effect is like scattering, which should suppress the tendency for ordering. Also, the measured ARPES band structure corresponds closely to theoretical results for a freestanding film. These considerations suggest that interaction with the substrate is not a key factor for the formation of the CDW phase.

We have grown TiTe_2 films on both doped and undoped SiC. We have not found any differences. This is perhaps not surprising, as the SiC substrate is separated from the TiTe_2 film by a bilayer of graphene.

4) As the monolayer of TiTe_2 exhibits the CDW phase, multilayer of TiTe_2 likely to have Kohn anomaly in their phonon band-structure, whereas unstable phonon mode appears in the monolayer case (EPL 115, 47001 (2016)).

Authors' response: The EPL paper presented a calculation for TiS_2 , where a phonon instability (imaginary frequency) is found at \bar{M} for the monolayer and a slight dip at \bar{M} is found for the bulk. In contrast, the electron-phonon interaction is weaker in TiTe_2 , as discussed above in the answer to Comment #1. We have calculated the phonon dispersion relations for a (1x1) single layer and bulk TiTe_2 and do not find any imaginary frequencies, as shown in the figure below. However, the lowering of the acoustic branch at \bar{M} in the single layer does indicate a tendency for (2x2) distortion. This discussion and the phonon dispersion relations are now included in the supplementary document (Section 7).

5) As the authors reported the formation of 2×2 super-structure for monolayer TiTe_2 in CDW phase, it would be interesting to show/report the strength of atomic displacements.

Authors' response: For TiSe_2 as a reference, the Ti and Se displacements are 0.08 and 0.02 Å, respectively (PRB Rapid Comm. 95, 201409 (2017)). Since the CDW transition in single-layer TiTe_2 is weaker with a lower transition temperature, the atomic displacements are expected to be correspondingly smaller. We do not have measurements for these values but intend to work on this problem in the future. For now, we are not ready to present any definitive statements. The effects of a weak CDW distortion are discussed in Section 4 of the supplementary document.

6) As the DFT-semi-local functional usually underestimate the bandgap, sometimes it is not promising in simulating the ARPES spectral weight accurately. In fact the their DFT calculation shows that 2×2 distorted super cell is not energetically favorable in the CDW phase. Such an ambiguity is needed to be addressed with more accurate calculations in their DFT calculations. Moreover the electronic band-structure would largely depend on the experimental growth environments, which needed to be discussed.

Authors' response: Since the states near the Fermi level are mainly of Ti $3d$ character, a good choice to include the correlation effect is to add a Hubbard U in the calculation, as has been done in our work. However, the calculation reported in EPL 115, 47001 (2016) showed that the LDA+ U scheme removed the phonon instability in monolayer TiS_2 , making the CDW phase unfavorable. For monolayer TiTe_2 , the energy difference between the undistorted and distorted structures is already extremely small. Including more correlation will not help stabilize the CDW phase.

There is only one structural phase of TiTe_2 , which is 1T. By varying the growth conditions, the films either grow well or not. We have not observed variations of the electronic band structure by changing the growth conditions, except when the film quality is bad, the measured band structure by ARPES is blurry and the surface is rough based on RHEED.

7) The authors have mentioned that due to metallic/semi-metallic behavior, exciton mechanism can be discarded. The exciton effect may have significant effect for such a monolayer thickness in which the confinement plays a crucial role. Previous study shows that exciton effect is significant even in semi-metallic graphene (Nature Nanotechnology 5, 32 (2009)). As the

energetic of the electronic structure is inconclusive, the role of exciton may provide some useful insights.

Authors' response: The excitonic insulator instability is strongly affected by screening effects. The bound states can form only if the number of carriers is small (Ref. 17). In our case, there is a large density of states near the Fermi level in single-layer, multi-layer, and bulk TiTe_2 (Fig. 3). Excitonic interactions would be screened out by the abundant charge carriers in this system. No "condensate" features have ever been observed by ARPES in TiTe_2 . The referee cited graphene as an example, as documented in *Nature Nanotechnology* 5, 32 (2009). However, the band gap of graphene is opened up by electrical gating in that study. In any case, the density of states at the Fermi level of graphene is generally very small. The situation is very different for TiTe_2 .

8) Although the authors have mentioned that distorted super-cell is not energetically favorable, it is very confusing to plot the DOS (Fig.S4) at different lattice constant. If the authors wanted to simulate the pressure effect, they need to discuss the effect in more details by discussing both the phonon and electronic structure.

Authors' response: The referee misunderstood our discussion. The DOS curves in Fig. S4 are actually computed with a fixed lattice constant assuming an amplitude of the Ti atomic displacement equal to 1% or 2% of the lattice constant. The CDW pattern is assumed to be the same as those in TiSe_2 , and the calculation is performed for the selected distortion amplitudes in order to compare with the STS results. We have revised the relevant sentence to make it clear; it now reads:

The distortion pattern is imposed on the lattice with a fixed lattice constant, assuming an amplitude of the Ti atomic displacement equal to 1% or 2% of the lattice constant.

Reviewers' Comments:

Reviewer #1:

Remarks to the Author:

The authors have addressed my questions and concerns with their substantial changes to their manuscript. I would recommend their work for publication.

Reviewer #2:

Remarks to the Author:

I think, in the present form, the paper can be published in Nature Communications.

Reviewer #3:

Remarks to the Author:

In the revised manuscript, the authors have addressed all the criticism raised by the referee. However, I mentioned in the previous report that the novelty of the manuscript is the finding of the CDW phase in monolayer of TiTe₂ only, whereas the CDW phase is absent in its bulk form. Therefore, I think that the authors need to address the origin of the CDW phase carefully or at least have to provide a clear direction to instigate the future research exploring the possible mechanism. The appearance of the CDW phase can be ascribed to several competing mechanisms such as Peierls instability (strong e-ph coupling), exciton insulator instability (many body interaction), Jahn-Teller distortion, or Fermi surface nesting (e-e correlation). However, their present theoretical and experimental results/discussions do not imply any of the mechanism for the CDW formation. Infact in the conclusion, the authors has mentioned that many-body interaction in the system might play significant role in the formation of the CDW phase. On the contrary, theoretical discussion indicates that both the electron-electron correlation and quasi-particle exciton formation do not play any significant role here, which seems to me confusing. Finally I am willing to recommend the manuscript for publication in Nature communication, once the authors will address the aforementioned concerns and others listed below.

1) The phonon dispersions of bulk and monolayer of TiTe₂ show that the kohn anomaly appears in the monolayer TiTe₂, whereas it absent in its bulk form. This indicate that the electron-phonon coupling gets enhanced, when the thickness is reduced from bulk to monolayer limit. Generally all the layered transition metal dichalcogenides exhibits a large eletron-phonon coupling. This fact might be an additional ingredient for the CDW formation. The authors made an attempt to address the issue in the response letter to the Referee#1, but their explanation is not quite convincing. Nevertheless, the kohn anomaly may turn out to be an unstable phonon mode, when we consider the fact that the DFT calculations are carried out properly (Phys. Rev. Lett. 112, 049701 (2014); EuroPhys.Lett. 115, 47001 (2016)). Otherwise, theoretical results do not provide any relevant explanations to the experimentally observed facts.

2) The authors have mentioned that exciton mechanism does not play any crucial role, however, previous study on TiSe₂ (very close electronic structure of TiTe₂) shows that it has a significant role [Phys. Rev. Lett. 98, 117007 (2007), Phys. Rev. Lett. 114, 086402 (2015)]. Therefore, I am not quite convinced with their explanation in the revised version.

Authors' response to reviewers' comments

Reviewer #1 (Remarks to the Author):

The authors have addressed my questions and concerns with their substantial changes to their manuscript. I would recommend their work for publication.

Authors' response: We thank the reviewer for his time and effort.

Reviewer #2 (Remarks to the Author):

I think, in the present form, the paper can be published in Nature Communications.

Authors' response: We thank the reviewer for his time and effort.

Reviewer #3 (Remarks to the Author):

In the revised manuscript, the authors have addressed all the criticism raised by the referee. However, I mentioned in the previous report that the novelty of the manuscript is the finding of the CDW phase in monolayer of TiTe₂ only, whereas the CDW phase is absent in its bulk form. Therefore, I think that the authors need to address the origin of the CDW phase carefully or at least have to provide a clear direction to instigate the future research exploring the possible mechanism. The appearance of the CDW phase can be ascribed to several competing mechanisms such as Peierls instability (strong e-ph coupling), exciton insulator instability (many body interaction), Jahn-Teller distortion, or Fermi surface nesting (e-e correlation). However, their present theoretical and experimental results/discussions do not imply any of the mechanism for the CDW formation. In fact in the conclusion, the authors has mentioned that many-body interaction in the system might play significant role in the formation of the CDW phase. On the contrary, theoretical discussion indicates that both the electron-electron correlation and quasi-particle exciton formation do not play any significant role here, which seems to me confusing. Finally I am willing to recommend the manuscript for publication in Nature communication, once the authors will address the aforementioned concerns and others listed below.

Authors' response: We thank the referee for acknowledging that we have addressed all of the referee comments in the previous round of review. He now recommends publication after we address some other concerns.

Regarding the mechanism of the surprising CDW transition in the single layer only, but not in thicker films and in the bulk, we do not really have any more stories to offer. As stated in the previous reply and also presented in the manuscript, we have honestly considered all available theories and models developed over the past decades, and none can explain the observation. Thus, the case is a mystery as stated in the manuscript, and the implication is that the current community-wide understanding of the underlying physics is missing something important. We believe that the discovery of the singular behavior of the single layer will greatly impact the field of CDW physics.

1) The phonon dispersions of bulk and monolayer of TiTe₂ show that the kohn anomaly appears in the monolayer TiTe₂, whereas it is absent in its bulk form. This indicates that the electron-phonon coupling gets enhanced, when the thickness is reduced from bulk to monolayer limit. Generally all the layered transition metal dichalcogenides exhibit a large electron-phonon coupling. This fact might be an additional ingredient for the CDW formation. The authors made an attempt to address the issue in the response letter to the Referee#1, but their explanation is not quite convincing. Nevertheless, the kohn anomaly may turn out to be an unstable phonon mode, when we consider the fact that the DFT calculations are carried out properly (Phys. Rev. Lett. 112, 049701 (2014); EuroPhys.Lett. 115, 47001 (2016)). Otherwise, theoretical results do not provide any relevant explanations to the experimentally observed facts.

Authors' response:

Referee 3 is quite correct that, in the first round of review, referee 1 asked about further details of electron-phonon coupling; in response, we provided theoretical phonon dispersion relations and expanded the discussion. Now, referee 1 has indicated his satisfaction, but referee 3 still thinks that our "explanation is not quite convincing." As emphasized in our manuscript, the current case is a mystery. Existing theories do not offer an explanation. We just honestly describe what we have found. To make absolutely sure that the message is clearly conveyed, we have added a sentence on page 8. The relevant part of the text now reads:

Calculated phonon dispersion relations for bulk and single-layer TiTe₂ (see Supplementary Document) show no imaginary frequencies that would correspond to structural instabilities. **The small dip in an acoustic branch at \bar{M} for the single layer might suggest an enhanced electron-phonon coupling compared to the bulk and thus a tendency for (2x2) distortion.**

The yellow highlighted part is the addition. A similar sentence was already contained in the supplementary document, but we have decided to repeat the message in the main text. We hope that the revision has removed any ambiguity.

Also, just to be sure that we have covered all the bases, we have performed phonon calculations for both the experimental and optimized lattice constants, as shown in the figure below. The differences are very slight; so, there are no surprises. Other effects including spin-orbit coupling and Hubbard U have also been tested. No phonon modes with imaginary frequencies are ever observed. This information is now included in the supplementary document and highlighted in the revised version.

2) The authors have mentioned that exciton mechanism does not play any crucial role, however, previous study on TiSe₂ (very close electronic structure of TiTe₂) shows that it has a significant role [Phys. Rev. Lett. 98, 117007 (2007), Phys. Rev. Lett. 114, 086402 (2015)]. Therefore, I am not quite convinced with their explanation in the revised version.

Authors' response: This point has already been addressed. As stated before, TiTe₂ is a very different case than TiSe₂. There is no band gap in TiTe₂, whereas there is a small band gap in single layer TiSe₂ in the normal phase (Ref. 7). The original paper by Walter Kohn on the exciton mechanism offers the following explanation. A semiconductor can host excitons. If the gap of the semiconductor is small and comparable in magnitude to the excitonic binding energy, the excitons could condense, resulting in a renormalization of the electronic structure around the gap. The basic premise for the excitonic insulator mechanism is a sufficiently small gap. TiSe₂ might be a potential candidate, but TiTe₂ is definitely not because there is a substantial overlap of the conduction and valence bands. The abundant charge carriers at the Fermi level would screen out any excitonic interactions. Experimentally, there is no evidence for an excitonic condensate that should give rise to an unusual many-body line shape. The spectral functions as determined by ARPES appear quite ordinary and are well described by the computed band structure. We believe that the case is quite clear.